# A Genetic Locus in *Elizabethkingia anophelis* Associated with Elevated Vancomycin Resistance and Multiple Antibiotic Reduced Susceptibility

**DOI:** 10.3390/antibiotics13010061

**Published:** 2024-01-08

**Authors:** William L. Johnson, Sushim Kumar Gupta, Suman Maharjan, Randy M. Morgenstein, Ainsley C. Nicholson, John R. McQuiston, John E. Gustafson

**Affiliations:** 1Department of Biochemistry and Molecular Biology, Oklahoma State University, Stillwater, OK 74074, USA; wiljohnson@salud.unm.edu (W.L.J.); sushim.gupta@okstate.edu (S.K.G.); 2Department of Microbiology and Molecular Genetics, Oklahoma State University, Stillwater, OK 74074, USA; suman.maharjan@okstate.edu (S.M.); randy.morgenstein@okstate.edu (R.M.M.); 3Special Bacteriology Reference Laboratory, Bacterial Special Pathogens Branch, Division of High-Consequence Pathogens and Pathology, Centers for Disease Control and Prevention, Atlanta, GA 30033, USA; agn0@cdc.gov (A.C.N.); zje8@cdc.gov (J.R.M.)

**Keywords:** *Elizabethkingia anophelis*, vancomycin selection, elevated vancomycin resistance, *padR* family transcriptional regulators, multiple antibiotic resistance

## Abstract

The Gram-negative *Elizabethkingia* express multiple antibiotic resistance and cause severe opportunistic infections. Vancomycin is commonly used to treat Gram-positive infections and has also been used to treat *Elizabethkingia* infections, even though Gram-negative organisms possess a vancomycin permeability barrier. *Elizabethkingia anophelis* appeared relatively vancomycin-susceptible and challenge with this drug led to morphological changes indicating cell lysis. In stark contrast, vancomycin growth challenge revealed that *E. anophelis* populations refractory to vancomycin emerged. In addition, *E. anophelis* vancomycin-selected mutants arose at high frequencies and demonstrated elevated vancomycin resistance and reduced susceptibility to other antimicrobials. All mutants possessed a SNP in a gene (*vsr1* = vancomycin-susceptibility regulator 1) encoding a PadR family transcriptional regulator located in the putative operon *vsr1-ORF551*, which is conserved in other *Elizabethkingia* spp as well. This is the first report linking a *padR* homologue (*vsr1*) to antimicrobial resistance in a Gram-negative organism. We provide evidence to support that *vsr1* acts as a negative regulator of *vsr1-ORF551* and that *vsr1-ORF551* upregulation is observed in vancomycin-selected mutants. Vancomycin-selected mutants also demonstrated reduced cell length indicating that cell wall synthesis is affected. ORF551 is a membrane-spanning protein with a small phage shock protein conserved domain. We hypothesize that since vancomycin-resistance is a function of membrane permeability in Gram-negative organisms, it is likely that the antimicrobial resistance mechanism in the vancomycin-selected mutants involves altered drug permeability.

## 1. Introduction

The Gram-negative *Elizabethkingia* are emerging opportunistic pathogens that cause severe disease (e.g., pneumonia, meningitis, and sepsis) in immunocompromised patients, the elderly, and neonates, that is associated with high mortality. To date, there are six characterized *Elizabethkingia* species that are known to cause human disease [1,2] and evidence suggests that most *Elizabethkingia* infections in humans are caused by *Elizaethkingia anophelis* [3]. Prior to a 2015–2016 *E. anophelis* community outbreak in the US Midwest involving 66 patients and 20 fatalities [4], outbreaks were primarily healthcare-related and often associated with water or a water source [5]. Three other species (*Elizabethkingia umeracha*, *Elizabethkingia argenteiflava* and an unnamed genomospecies) not yet associated with human infection were isolated from environmental and agricultural sources [6,7,8].

The *Elizabethkingia* exhibit intrinsic multiple antimicrobial resistance, which contributes to the high mortality attributed to infections caused by these organisms. Experimental research on the mechanisms of antimicrobial resistance in *Elizabethkingia* have focused on beta-lactamase mediated resistance [9] and fluoroquinolone resistance [10]. A unique aspect of these organisms is that they harbor multiple copies of putative beta-lactamase genes on their chromosome [11]. The *Elizabethkingia* also harbor multiple operons encoding putative Resistance-Nodulation-Division (RND) family intrinsic multidrug trans-envelope efflux pumps [12] which have been linked to clinically-relevant antimicrobial resistance [13].

The primary mechanism of action of the glycopeptide antibiotic vancomycin is the inhibition of peptidoglycan biosynthesis. Vancomycin binds to the terminal D-Ala-D-Ala residues of emerging muramyl-peptides on the membrane surface and prevents the linkage of new peptidoglycan monomers to the existing cell wall [14]. Vancomycin is used almost exclusively to treat Gram-positive infections [15] but it has also been used to treat *Elizabethkingia* infections with variable success [16]. The use of vancomycin to treat *Elizabethkingia* infections is highly unusual since Gram-negative organisms should be vancomycin-resistant due to the outer membrane permeability barrier and the molecular size exclusion (<600 Da and vancomycin is 1449 Da) of the aqueous filled outer membrane porins [17]. Drug efflux does not appear to play a role with resistance to vancomycin in Gram-negative organisms [17]. 

In the absence of Clinical and Laboratory Standards Institute (CLSI) guidelines for the *Elizabethkingia*, MIC breakpoints and Kirby Bauer zones of inhibition from the Gram-positive organism *Staphylococcus aureus* have been used to evaluate vancomycin susceptibility in *Elizabethkingia* [16], but several lines of evidence suggested that this is a mistake. Chiu et al., 2021 [18], provided evidence that disk diffusion and the E-test should not be employed to determine vancomycin susceptibility for *E. anophelis*. While most Gram-negative organisms exhibit high vancomycin MICs (64 to >1000 mg/L) [17,19,20], the *Elizabethkingia* demonstrate vancomycin MICs in the range of 1 to 64 mg/L [16,21,22,23] (this study). One study with 167 clinical *Elizabethkingia* isolates reported that 95.8% of *E. anopheles* and 100% of *Elizabethkingia meningoseptica* demonstrated intermediate vancomycin susceptibility [21]. Furthermore, it was reported that 108 *Elizabethkingia* spp. isolates demonstrated vancomycin resistance [22] while another study with 84 *E. anophelis* isolates also reported these strains were resistant to vancomycin as determined with agar dilution assays [18]. Collectively these findings support the notion that *Elizabethkingia* should be classified as demonstrating intermediate to full resistance to vancomycin. 

To date there are no in vitro studies of the effects of vancomycin on *Elizabethkingia* yet considering the controversial use of this drug to treat infections caused by these Gram-negative organisms, thus such a study was warranted. The objectives of our research were to examine the effects of vancomycin on growth and the cell structure of *Elizabethkingia*, determine if vancomycin selection readily increases *Elizabethkingia* vancomycin-resistance, and determine the identity of mutations that alter vancomycin resistance in these organisms. This study provides more information on the nature of intrinsic antimicrobial resistance in these opportunistic pathogens.

## 2. Results

### 2.1. Vancomycin Susceptibility, Live Microscopy, and Vancomycin Survival Assays 

The vancomycin MIC and MBC of *E. anophelis* R26 are shown in Table 1. Compared to other Gram-negative organisms [19], the vancomycin MIC and MBC for *E. anophelis* R26 appeared relatively low (Table 1), yet were in general, higher than the concentrations vancomycin can reach in certain tissues [24]. A representative sample of *E. anophelis* exposed to 1.5 X the vancomycin MIC for 4 h is shown in Figure 1. Exposure gave rise to a large percentage (78.44%) of lightly colored “empty” cells (white arrows in Figure 1) and we also noted numerous cells that demonstrated blebbing of the cell membrane (black arrow in Figure 1) which is indicative of bacterial cell wall degradation. Following 2 h of incubation, *E. anophelis* CFUs/mL began to decline precipitously in the 1.5 X MIC and the MBC culture up to 8 h (Figure 2). These cultures then rebounded to reach similar CFUs/mL as the control culture by the 24 h timepoint (Figure 2).

### 2.2. Isolation of Vancomycin-Selected Mutants

In studies with *Escherichia coli* and *Staphylococcus aureus*, the mutation frequency for antibiotic resistance was reported to be between 10^−5^ and 10^−9^ [25,26,27], which means that one cell in population of 10^5^ to 10^9^ cells possessed spontaneous mutation(s) that granted antibiotic resistance. Vancomycin-selected mutants of *E. anophelis* appeared on media containing 16 mg/L vancomycin at a frequency of 10^−4^, which represented an unusually high frequency (Table 1). Three randomly chosen vancomycin-selected R26 mutants (R26VS1, R26VS2 and R26VS3) demonstrated higher vancomycin MICs and MBCs compared to parent strain R26 (Table 1). Gradient plate analysis (Table 2) also revealed that all R26 vancomycin-selected mutants demonstrated reduced susceptibility to ciprofloxacin, clindamycin, and rifampin. Under the microscope, we noted that the R26 parent cells had a longer rod length (2.37 ± 0.59 mm, n = 1652) compared to R26VS1 (2.09 mm ± 0.49 mm, n = 3417, *p* < 0.001) and R26VS2 (2.11 ± 0.47 mm, n = 2138, *p* < 0.001). 

### 2.3. Mutations Associated with Vancomycin Resistance

Genome sequencing revealed that all three randomly picked R26 vancomycin-selected mutants possess a single cytosine insertion in a PadR-family helix-turn-helix transcriptional regulator gene [28,29,30,31] encoding a 112 aa product (WP_009089502) we have termed the “vancomycin-susceptibility regulator 1” or *vsr1* (Figure 3). *vsr1* overlaps by 7 bps and forms a putative bicistronic operon with *ORF551* (Figure 3) which encodes a 551 aa protein (WP_009089500.1) that possesses five transmembrane-spanning regions and an internal small phage shock protein (58 aa)-conserved domain. Bioinformatic analysis of the upstream *vsr1* promoter region did not reveal a PadR binding consensus site, but it did uncover a putative binding site for a helix-turn-helix SoxS regulatory protein, a RpoD sigma factor binding site, and -35 and -10 consensus sequences (Figure 3).

The insertion of the cytosine in *vsr1* (Figure 3) resulted in a frameshift mutation that caused three amino acid substitutions (R^75^ → T^75^, Y^77^ → I77, and Y^78^ → L^78^) and introduced a premature stop codon at aa position 79 in Vsr1 (Figure 4). This nonsense mutation leads to the deletion of the C-terminal 34 amino acids in Vsr1 which contains an important PadR dimerization domain [32]. To determine conserved motifs within *Elizabethkingia* PadR homologues, the amino acid sequences for 293 *Elizabethkingia* PadRs were downloaded from NCBI and compared via multiple alignments with the program mafft [33], and the data was visualized using WebLogo [34]. This analysis indicated that all *Elizabethkingia* PadRs share 41 conserved amino acids (or demonstrate 35% amino acid identity) with 100% conservation of a RKYY motif that is the beginning of the deletion observed in the vancomycin selected *E. anophelis* mutants (Figure 4). 

**Table 1 antibiotics-13-00061-t001:** Strains utilized in study.

Strain	Parent Strain	Vancomycin Selection Concentration (mg/L)	Mutation Frequency	Vancomycin MIC (mg/L)	Vancomycin MBC (mg/L)	Ref.
*E. anophelis* R26				8	16	[34]
R26VS1	R26	16	4.33 × 10^−4^	128	>256	This study
R26VS2	R26	16	4.33 × 10^−4^	64	128	This study
R26VS3	R26	16	4.33 × 10^−4^	64	128	This study

**Table 2 antibiotics-13-00061-t002:** Gradient plate antibiotic susceptibility analysis.

Strain	Ciprofloxacin 0 → 0.5 mg/L	Clindamycin 0 → 1 mg/L	Rifampin 0 → 0.25 mg/L	Vancomycin 0 → 64 mg/L
R26	3.67 ± 0.33 ^A^	31.00 ± 1.15 ^A^	41.33 ± 1.76 ^A^	6.33 ± 0.67
R26 VS1	7.67 ± 0.67 ^BC^	65.67 ± 1.45 ^B^	63.67 ± 2.60 ^B^	90.00 ± 0.00
R26 VS2	7.00 ± 0.58 ^C^	52.67 ± 1.20 ^C^	70.33 ± 2.03 ^B^	89.00 ± 1.00
R26 VS3	9.67 ± 0.33 ^B^	61.67 ± 2.19 ^B^	80.33 ± 1.45 ^C^	90.00 ± 0.00

One-way ANOVA with groups (A, B, and or C) displaying significant differences in susceptibility to the individual antibiotics ciprofloxacin, clindamycin, and rifampin (*p* < 0.05). Because strains grew to the top of the vancomycin gradient, statistical analyses could not be performed.

We then completed a phylogeny of the PadRs found in the genomes of five *Elizabethkingia*-type strains compared to three PadRs (PadR1, PadR2 and PadR3) found within a high quality *Burkholderia cepacia* complete genome (Figure 5). This analysis revealed the presence of two structurally distinct PadR subfamilies (PadR1 and PadR2) in the *Elizabethkingia* and Vsr1 belongs to the PadR1 subfamily. The PadR1 subfamily is characterized by the separation of the DNA binding domain in the N-terminal from the dimerization domain in the C-terminal region. An *ORF551* gene lies upstream of all *padR1s* in all five *Elizabethkingia* genomes examined (see materials methods Section 4.4). The gene products encoded for by these genes exhibit 95.54 to 100% aa identity for Vsr1 and 89.67% to 100% aa identity for ORF155, demonstrating that the putative *vsr1-ORF551* operon is conserved in the *Elizabethkingia*. 

### 2.4. qPCR Analysis of vrs1 Expression

Compared to parent strain R26, the expression levels of both *vsr1* and *ORF551* were increased in R26VS1 (224.4 ± 0.3 and 99.73 ± 0.1, respectively) and R26VS2 (272.4 ± 0.7 and 80.44 ± 0.1, respectively; N = 3, ±standard error) in drug-free cultures. These genes were also upregulated following vancomycin induction in R26VS1 and R26VS2 (Table 3), and only *vsr1* was upregulated by vancomycin induction in R26 (Table 3).

## 3. Discussion

The vancomycin MIC and MBC for *E. anophelis* appeared to be relatively low for Gram-negative organisms and this organism responded to a growth inhibitory vancomycin concentration in a fashion that indicates the inhibition of cell wall synthesis. The formation of empty R26 cells with associated membrane blebbing following vancomycin challenge was similar to those reported by Huang et al. [35] who challenged vancomycin-susceptible *E. coli* mutants with vancomycin and noted similar abnormalities. The vancomycin growth survival challenge, however, indicated that *E. anophelis* cell populations were selected that were refractory to the action of vancomycin and *E. anophelis* vancomycin-selected mutants appeared on media containing a growth inhibitory vancomycin concentration at unusually high mutation frequencies. These findings demonstrate the ineffectiveness of vancomycin action against a population of *E. anophelis* and lends support to studies demonstrating that the *Elizabethkingia* display intermediate to full resistance to vancomycin [18,21,22]. We provide additional evidence that the vancomycin-selected mutants also demonstrated reduced susceptibility to other antimicrobials. The relative ease of selection for elevated vancomycin resistance and multiple antibiotic susceptibility mechanism should raise concern since vancomycin has been used to treat *Elizabethkingia* infections.

All three randomly chosen vancomycin-selected R26 mutants examined had the same mutation within *vsr1*, which encodes a PadR homologue that is located in the putative bicistronic operon *vsr1*-*ORF551*, an operon that is present in all *Elizabethkinigia* analyzed. Sequence analysis of 293 *Elizabethkingia* putative PadR proteins revealed these proteins exhibit a great deal of amino acid conservation and a highly conserved RKYY motif in the C-terminal domain of *Elizabethkingia* PadRs, which represents the beginning of the deletion in Vsr1 observed in all vancomycin-selected mutants of *E. anophelis*. Based on this bioinformatic analysis, we hypothesized that the truncated Vsr1 missing the C-terminal 34 amino acids and the conserved RKYY motif in vancomycin selected mutants cannot bind DNA and Vsr1 activity is compromised in these mutants. Our data also demonstrated that the *vsr1-ORF551* operon was upregulated in vancomycin-selected mutants, suggesting that Vsr1 acts in one way or another as a repressor of the *vsr1-ORF551* operon. However, no PadR binding site was identified upstream of the *vsr1* start codon which was not wholly unexpected given the diversity of PadR regulators and therefore the sequences they bind [28,29,30,31], and the documented difficulties in promoter prediction for less well-studied organisms. Analysis of the promoter region upstream of *vsr1-ORF551* did however identify a potential binding site for another helix-turn-helix SoxS type regulatory protein, and it should be noted that *soxS* genes have been implicated to play a role in the regulation of genes associated with intrinsic multiple antimicrobial resistance in multiple Gram-negative bacterial species [36,37,38,39]. 

The Vsr1 and ORF155 proteins of *Elizabethkingia* spp. are highly conserved and the genes encoding these proteins demonstrated synteny among the species analyzed. Additional experimentation is however required to determine if *vsr1-ORF551* operons are associated with intrinsic antimicrobial resistance in other *Elizabethkingia*. In addition, we identified two distinct *Elizabethkingia* PadR subfamilies suggesting that these diverged gene families likely provide distinct functions to the cell.

PadR homologues in Gram-positive bacteria control genes encoding multidrug efflux pumps [28,29,30,31] and a *padR* operon in *Streptococcus pneumoniae* expressed several membrane proteins with unknown functions that control vancomycin tolerance [40]. PadR homologues have a structure that is similar to the MarR family of proteins (which includes SoxS), which have been reported to control intrinsic multiple antimicrobial resistance in the *Enterobacteriaceae*. In *E. coli*, MarR regulates the expression of MarA, which controls the production and activity of outer membrane porins and RND efflux pumps such as AcrAB-TolC, leading to a reduction in antimicrobial accumulation [41,42,43]. To the best of our knowledge, this is the first description of a PadR homologue playing a role in antimicrobial resistance in a Gram-negative organism.

The vancomycin-selected mutants were shorter than the parent strain and since cell morphology is maintained by peptidoglycan structure [44], we suggest that cell wall biosynthesis is affected in these mutants. *ORF551* encodes a protein with membrane spanning regions and possesses a small phage shock protein-conserved domain. In *Escherichia coli*, it has been proposed that the phage shock protein system can detect and mitigate issues that affect inner membrane permeability [45]. Since Gram-negative organisms do not efflux vancomycin and are reported to be vancomycin-resistant due to an inherent outer membrane permeability barrier [17], we hypothesize that the vancomycin-selected mechanism affects membrane permeability in *E. anophelis*.

We now intend to determine the effects of the inactivation and complementation of the *vsr1*-*ORF551* operon on intrinsic antimicrobial resistance and antimicrobial accumulation. 

## 4. Materials and Methods

### 4.1. Bacterial Strains, Growth Conditions, Vancomycin-Selected Mutants, Antibiotic Susceptibility Testing, and Live Cell Microscopy

The type strain *Elizabethkingia anophelis* R26 isolated from the midgut of *Anopheles gambiae* [46] was used for this study. A complete list of bacterial strains used in this study can be found in Table 1. All freezer stocks were maintained in heart infusion broth (HIB) containing 20% *v*/*v* glycerol (final concentration) at −80 °C. Working stocks were maintained on heart infusion agar (HIA; Remel, San Diego, CA, USA) supplemented with 5% defibrinated rabbit blood (Hemostat Laboratories, Dixon, CA, USA). All overnight cultures were prepared by inoculating a single colony into HIB or Mueller-Hinton (MHB) broth, followed by overnight incubation (37 °C, 200 rpm, 18 h). All chemicals and antibiotics were purchased from MilliporeSigma (St. Louis, MO, USA).

Vancomycin-selected mutants of *E. anophelis* R26 were isolated by plating diluted HIB overnight cultures onto HIA plates supplemented with 16 mg/L of vancomycin (MilliporeSigma, St. Louis, MO, USA). Following overnight incubation (37 °C) single isolated colonies were picked, passaged three times on HIA, and HIB glycerol freezer archive stocks were prepared. Mutation frequencies (Table 1) were defined as the proportion of the colonies growing on vancomycin selection plates to the total viable colony count.

Vancomycin MIC and MBC concentrations were determined by broth dilution following standard CLSI guidelines [47] and relative antibiotic susceptibility was compared utilizing the antimicrobial gradient plate analysis as described previously [48,49]. Distances grown on antimicrobial gradient plates (Table 1) were analyzed by one-way ANOVA with groups displaying significant differences (*p* < 0.05) and subsequently differentiated using Tukey’s Honestly Significant Differences post hoc testing [50]. All analysis was performed using JMP Pro (version 14; SAS Institute Inc., Carey, NC).

Vancomycin growth survival assays were performed in 50 mL flasks containing MHB cultures initiated with overnight culture (beginning OD600nm = 0.01) containing no addition or vancomycin (1.5 X MIC and MBC). These flasks were then incubated with shaking (200 rpm, 37 °C) and surviving 

CFUs/mL were determined over time by plating dilutions onto Mueller-Hinton agar (MHA) followed by overnight incubation (37 °C).

For live cell microscopy, overnight cultures were diluted in fresh MHB to reach an OD_600nm_ = 0.01 and incubated for 3 h (37 °C, 200 rpm). Following incubation, vancomycin was added to a final concentration of 1.5 X the MIC for each isolate and a 1 µL aliquot was transferred to a sterile 1% agar pad at 25 °C for visualization. Vancomycin-challenged cultures were then incubated for 4 h (37 °C, 200 rpm), with 1 µL aliquots removed for imaging at 2 h and 4 h post-challenge. Phase contrast images were collected on a NikonNi-E epifluorescent microscope equipped with a 100X/1.45 NA objective (Nikon, Tokyo, Japan), Zyla 4.2 plus cooled sCMOS camera (Andor, Belfast, Northern Ireland), and NIS Elements software (Nikon). 

### 4.2. Whole Genome Sequencing and Identification of Mutations Associated with Enhanced Vancomycin Resistance

Upon arrival at CDC, strains were grown on HIA (Difco, Tokyo, Japan) supplemented with 5% rabbit blood (Hemostat Laboratories) at 35 °C. Genomic DNA was extracted using the CTAB protocol provided by the Department of Energy’s Joint Genome Institute [51]. *E. anophelis* vancomycin-selected mutant libraries were prepared using the NexteraTM DNA Flex kit according to manufacturer’s instructions, and genomes were sequenced using a 2 × 150 paired end protocol using an Illumina iSeq 100 Sequencing System (Illumina, Inc., San Diego, CA, USA).

Paired raw reads were trimmed to remove adapter sequences and for quality control using a quality threshold of 0.02 and 0 allowable ambiguous nucleotides. Trimmed reads were then mapped to the complete R26 genome using the default options and the consensus sequence for each isolate was extracted. All reported mutations were verified by inspection of the raw reads. All trimming and mapping steps were performed using CLC Genomics Workbench v11.0.1. Consensus sequences were annotated using the Rapid Annotations Using Subsystems Technology (RAST) server [52]. Regulatory elements were predicted using the BPROM program [53] while the identity and putative functional domains of hypothetical proteins were investigated using nucleotide and protein Basic Local Alignment Search Tool (BLAST) [54].

### 4.3. Quantitative Real-Time qPCR Analysis

Overnight HIB cultures (in triplicate) were used to inoculate (0.1% *v*/*v* inoculum) fresh HIB which was incubated (37 °C, 200 rpm) until an OD_600nm_ = 0.5. These cultures were then divided into equal portions, and one portion of the culture was induced with 4 mg/L vancomycin. All cultures were then incubated (37 °C, 200 rpm) and the cells were harvested after 30 min growth. Total RNA was isolated utilizing Trizol (Invitrogen, ThermoFisher Scientific, Waltham, MA, USA) according to the manufacturer’s instructions. The isolated RNA was then solubilized in the RNase-free water and treated with DNA-free(tm) (Ambion, Life Technologies, Carlsbad, CA, USA) according to the manufacturer’s instructions. RNA quantity and quality were then measured on a NanoDrop ND-1000 spectrophotometer (Thermo Fisher Scientific, Wilmington, DE, USA). cDNA synthesis was carried out with isolated RNA (25 ng) using the High-Capacity cDNA Reverse Transcription Kit according to the manufacturer’s proto-cols (Applied Biosystems™, Thermo Fisher Scientific, Vilnius, Lithuania). The cDNA samples were then diluted five-fold and the relative quantification of the target genes in triplicate was carried out using the LightCycler^®^ 96 real-time PCR system (Roche Diagnostics, Mannheim, Germany) and iQ SYBR^®^ Green Supermix as per the manufacturer’s recommended protocols (Bio-Rad, Hercules, CA, USA). Gene expression was then normalized using species-specific *rpoB* primers, expression levels were calculated using the 2^−ΔΔCT^ method [55], and results are presented as the means and standard errors of the data. All the primers used for RT-qPCR are shown in Table 4. 

### 4.4. Promoter, Structural Sequence and Phylogenetic Analyses

All *vsr1* promoter sequences and potential DNA binding protein binding sites were identified with a previously described program [53]. Amino acid sequences for 293 putative *Elizabethkingia* PadR genes were downloaded from NCBI, aligned by mafft [56], and then visualized using WebLogo [34]. To further investigate the differences between the two PadR families in *Elizabethkingia*, PadR amino acid sequences were downloaded from the NCBI RefSeq for each *Elizabethkingia* species (*E. anophelis* R26, accession # GCF_002023665.2; *E. bruuniana* FDAARGOS_1031, accession # GCF_016599835.1; *E. meningoseptica* G4120, accession # GCF_002022145.1; *E. occulta* G4070, accession # GCF_002023715.1; and *E. ursingii* CSID_3000516135, accession # GCF_002023405.1), along with three PadR sequences from *Burkholderia cepacia* (strain BC16, accession # GCF_009586235.1) to serve as a comparator. Amino acid sequences were aligned by mafft, and maximum likelihood phylogenies were created by IQ-Tree using the Q.pfam + I model, with the -bb and -alrt options set for 10,000 bootstraps each. 

## 5. Conclusions

In the laboratory, vancomycin was readily selected for elevated vancomycin resistance, intrinsic multiple antimicrobial reduced susceptibility, and altered cell wall morphology in *E. anopheles*. A mutation within a *padR* homologue (*vsr1*) within a putative bicistronic operon *vsr1-ORF551* and *vsr1-ORF551* upregulation, was observed in vancomycin-selected mutants. The *vsr1-ORF551* operon is conserved in the *Elizabethkingia* and we identified two PadR genes within genomes of this genus. This work represents the first time a *padR* homologue and a gene like *ORF551* have been associated with intrinsic antimicrobial resistance in a Gram-negative organism. Overall, this work adds to the knowledge of mutation-mediated intrinsic antimicrobial resistance mechanisms of the *Elizabethkingia* and adds credence to the literature that demonstrated that these organisms are vancoycin-resistant.

## Figures and Tables

**Figure 1 antibiotics-13-00061-f001:**
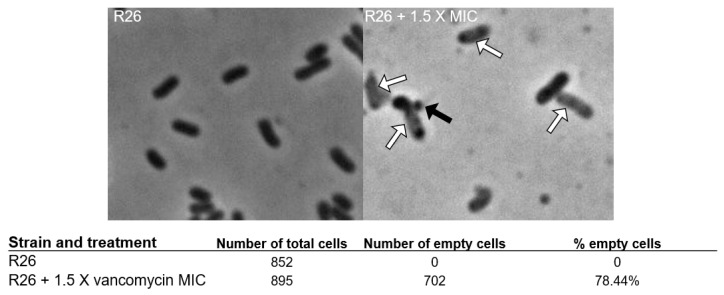
Representative light microscopy pictures of R26 controls and R26 challenged with 1.5 X the vancomycin MIC. Black arrow points to representative membrane bleb, white arrows point to empty cells.

**Figure 2 antibiotics-13-00061-f002:**
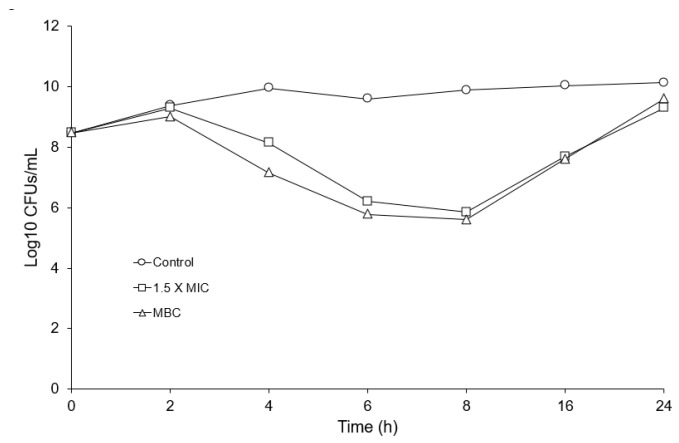
Vancomycin growth survival assay of R26 challenged with 1.5 X MIC and the vancomycin MBC, showing surviving Log10 CFUs/mL over time.

**Figure 3 antibiotics-13-00061-f003:**
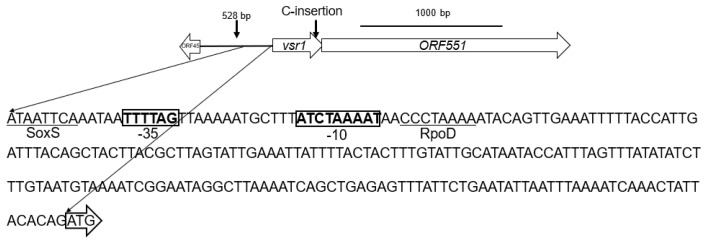
*E. anopheles vsr1-ORF551* operon showing location of cytosine (C) insertion at the end of *vsr1* in R26 vancomycin-selected mutants and the 233 base pair promoter sequence upstream of *vsr1.* The *vsr1* start codon is surrounded by an open arrow, the SoxS and RpoD sigma factor consensus binding sites are indicated (underlined) as are −35 and −10 regions (bolded in boxes) in the *vsr1* promoter region.

**Figure 4 antibiotics-13-00061-f004:**
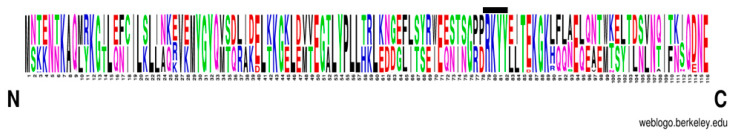
Sequence logo produced by alignment of 293 PadR sequences from all known *Elizabethkingia* species revealing highly conserved areas (positions represented by single large letters). The conserved RKYY motif where the truncation of Vsr1 occurs in all vancomycin-selected mutants is indicated by the black bar above amino acids.

**Figure 5 antibiotics-13-00061-f005:**
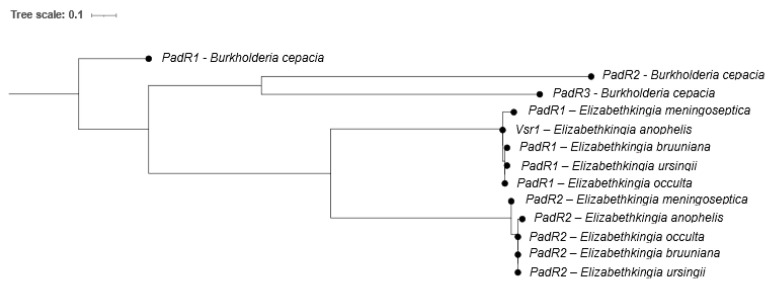
Maximum likelihood phylogeny of PadR amino acid sequences from *Elizabethkingia* spp. PadR genes from *Burkholderia cepacia* BC16 are included as an outgroup.

**Table 3 antibiotics-13-00061-t003:** Relative gene expression in vancomycin-induced cultures compared to uninduced.

Strain Gene	R26	R26VS1	R26VS3
*vsr1*	1.95 ± 0.18	172.4 ± 0.17	199.46 ± 0.05
*ORF551*	0.97 ± 0.12	70.03 ± 0.25	61.81 ± 0.19

N = 3, ±standard error.

**Table 4 antibiotics-13-00061-t004:** List of primers used in qRT-PCR analysis.

Target Gene	Primer Name	Sequence
*vsr1*	*Ea-vsr1-F*	5′-GAATACCAAAGCGCAAATG-3′
	*Ea-vsr1-R*	5′-ACTTGTAGACTCTTCCCAA-3′
*orf551*	*Ea-orf551-F*	5′-CGTCGTTCTATGGAGCCTGA-3′
	*Ea-orf551-R*	5′-CGGTGTACCGATAAGGGCAA-3′
*rpoB*	*Ea-rpoB-F*	5′-TGTACTGACCCGGAACATGA-3′
	*Ea-rpoB-R*	5′-CGGTGAACGGTGTAACTGAG-3′

## Data Availability

Sequence data for strains R26-VER1, R26-VER2, and R26-VER3 have been deposited to NCBI BioProjects PRJNA875105, PRJNA875110, and PRJNA875117 respectively, and released with the accession numbers JANZKF000000000, JANZKG000000000, and JANZKH000000000.

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
