# Peer review of "A Genetic Locus in Elizabethkingia anophelis Associated with Elevated Vancomycin Resistance and Multiple Antibiotic Reduced Susceptibility"

_antibiotics, 2024, doi:10.3390/antibiotics13010061_

Round 1

Reviewer 1 Report

Comments and Suggestions for Authors

In the submitted manuscript the authors address the issue of antibiotic resistance in an emerging opportunistic pathogen, namely Elizabethkingia anophelis, a Gram-negative bacterium resistant to vancomycin. By growing in vitro a reference Elizabethkingia anophelis strain in the presence of vancomycin at MBC (minimum bactericidal concentration), the authors selected three vancomycin-resistant mutants, all having a single cytosine insertion in the gene vsr1 (“vancomycin-susceptibility regulator 1”) of the PadR transcriptional regulators family. As a consequence, the mutated vsr1 protein showed three amino acid substitutions and a premature stop codon. The authors hypothesize these mutations negatively affected the activity of the putative vsr1 protein. PadR homologues were found in Gram-positive bacteria, where they are involved in the control of genes encoding multidrug efflux pumps. The principal novelty of the submitted manuscript is the finding that such a homologue was found to have a role in vancomycin resistance in the Gram-negative Elizabethkingia genus.

The study meets the aims and scope of the journal. It is well written and the methods are correctly applied.

Data are clearly presented.

However, there are some minor points to be corrected:

lines 129 and 132: please, specify the meaning of the codes WP_009089502 and WP_009089500.1, respectively;

lines 141-154: The relative gene expression of the vsr1 and ORF551 genes in the mutants is reported in the text as in a “drug free” condition (lines 141-143), and in the text (lines 143-145) and in Table 3 (lines 146-154) as “compared to uninduced” (line 146). In my opinion, such a presentation could raise confusion. Could you improve this part? Maybe by putting un-induced and induced data in the same table?

Author Response

Reviewer 1

Comment 1. : In the submitted manuscript the authors address the issue of antibiotic resistance in an emerging opportunistic pathogen, namely Elizabethkingia anophelis, a Gram-negative bacterium resistant to vancomycin. By growing in vitro a reference Elizabethkingia anophelis strain in the presence of vancomycin at MBC (minimum bactericidal concentration), the authors selected three vancomycin-resistant mutants, all having a single cytosine insertion in the gene vsr1 (“vancomycin-susceptibility regulator 1”) of the PadR transcriptional regulators family. As a consequence, the mutated vsr1 protein showed three amino acid substitutions and a premature stop codon. The authors hypothesize these mutations negatively affected the activity of the putative vsr1 protein. PadR homologues were found in Gram-positive bacteria, where they are involved in the control of genes encoding multidrug efflux pumps. The principal novelty of the submitted manuscript is the finding that such a homologue was found to have a role in vancomycin resistance in the Gram-negative Elizabethkingia genus.

Response: We thank the reviewer for this positive comment.

Comment 2. The study meets the aims and scope of the journal. It is well written and the methods are correctly applied.

Response: We thank the reviewer for this positive comment.

Comment 3. Data are clearly presented.

Response: We thank the reviewer for this positive comment.

Comment 4. lines 129 and 132: please, specify the meaning of the codes WP_009089502 and WP_009089500.1, respectively;

Response: These are the protein ID numbers for Vsr1 and ORF551 respectively, so that readers can look at the proteins themselves in the NCBI databases. This is commonly added to manuscripts.

Comment 5. lines 141-154: The relative gene expression of the vsr1 and ORF551 genes in the mutants is reported in the text as in a “drug free” condition (lines 141-143), and in the text (lines 143-145) and in Table 3 (lines 146-154) as “compared to uninduced” (line 146). In my opinion, such a presentation could raise confusion. Could you improve this part? Maybe by putting un-induced and induced data in the same table?

Response: We thank the reviewer for their comment, however we feel as presented the results are clear. We chose to show the induced data in a table format since only vrs1 was upregulated in R26 by vancomycin induction and we thought this would allow the reader to scrutinize this induced data better.

Reviewer 2 Report

Comments and Suggestions for Authors

It is an interesting article on vancomycin susceptibility of Elizabethkingia anopheles to Vancomycin.

I would recommend giving a little bit more background on the pathogenesis of the bacterium including if where it is commonly found and how it is acquired. I had never heard of it.

I was not very clear from the abstract, why vancomycin was even prescribed for Elizabethkingsia infection anyway since Gram negative bacteria are naturally resistant to vancomycin. A little bit more comparison with the efflux pumps present in other Gram negative bacteria and Elizabethkingsia would be very helpful.

Comments on the Quality of English Language

The quality of English overall is good. There are some grammatical errors that can be corrected by going over the manuscript carefully.

Author Response

Report 2:

It is an interesting article on vancomycin susceptibility of Elizabethkingia anopheles to Vancomycin.

Response: We thank the reviewer for this positive comment.

Comment 1.  I would recommend giving a little bit more background on the pathogenesis of the bacterium including if where it is commonly found and how it is acquired. I had never heard of it.

Response: We thank the reviewer for their comment, however we feel as the introduction as written along with the references is enough information to support the research presented and provides the reader with references they can seek out should they want more information on the subject matter.

Comment 2. I was not very clear from the abstract, why vancomycin was even prescribed for Elizabethkingsia infection anyway since Gram negative bacteria are naturally resistant to vancomycin.

Response: We thank the reviewer for their comment, the fact that there is literature that suggesting that vancomycin can be used to treat infections caused by these organisms is the very reason why we conducted these experiments. While our work is not a clinical trial, it does suggest there is a major problem that results from exposing Elizabethkinigia populations to vancomycin at least in vitro. The references in the introduction provide a great deal of background material for the reader to access.

Comment 3. A little bit more comparison with the efflux pumps present in other Gram negative bacteria and Elizabethkingsia would be very helpful.

Response: We thank the reviewer for their comment, and we point the reader to reference 47 in the revised manuscript where our laboratory provides a full description of putative efllux pumps found in the Elizabethkingia genome. We do not focus on efflux pumps except to say they are present in Elizabethkingia, because while they are major mediators of antibiotic resistance, they are not thought to play a role in susceptibility to vancomycin in Gram-negatives. Rather we propose a hypothesis that our mutation and operon upregulation leads to reduced membrane permeability of vancomycin, leading to increased vancomycin resistance which is currently under investigation.
